# Selectins: An Important Family of Glycan-Binding Cell Adhesion Molecules in Ovarian Cancer

**DOI:** 10.3390/cancers12082238

**Published:** 2020-08-10

**Authors:** Ayon A. Hassan, Margarita Artemenko, Maggie K.S. Tang, Alice S.T. Wong

**Affiliations:** School of Biological Sciences, University of Hong Kong, Pokfulam Road, Hong Kong, China; h1259275@connect.hku.hk (A.A.H.); mart2602@connect.hku.hk (M.A.)

**Keywords:** ovarian carcinoma, peritoneal metastasis, selectin, glycan, shear stress

## Abstract

Ovarian cancer is the most lethal gynecological malignancy worldwide. Unlike most other tumor types that metastasize via the vasculature, ovarian cancer metastasizes predominantly via the transcoelomic route within the peritoneal cavity. As cancer metastasis accounts for the majority of deaths, there is an urge to better understand its determinants. In the peritoneal cavity, tumor-mesothelial adhesion is an important step for cancer dissemination. Selectins are glycan-binding molecules that facilitate early steps of this adhesion cascade by mediating heterotypic cell-cell interaction under hydrodynamic flow. Here, we review the function and regulation of selectins in peritoneal carcinomatosis of ovarian cancer, and highlight how dysregulation of selectin ligand biogenesis affects disease outcome. Further, we will introduce the latest tools in studying selectin-glycan interaction. Finally, an overview of potential therapeutic intervention points that may lead to the development of efficacious therapies for ovarian cancer is provided.

## 1. Introduction

Ovarian cancer is the most lethal gynecological cancer in terms of the case-to-fatality ratio [1]. The majority of patients are diagnosed at an advanced stage with widespread peritoneal dissemination and malignant ascites. The metastatic spread is a multi-step process that involves (1) detachment from the primary tumor, (2) dissemination, and (3) peritoneal implantation [2]. Due to non-specific symptoms and late presentation of the disease, the 10-year survival remains at around 30% [3]. Thus, there is an urge to better understand the underlying molecular mechanisms of ovarian cancer metastasis.

In ovarian cancer, due to the lack of an anatomical barrier, cancer cells detach from the primary tumor and metastasize through the peritoneal cavity [4]. The formation of malignant ascites provides a complex microenvironment that is enriched with various biochemical (e.g., cytokines and growth factors) and biomechanical (e.g., stiffness and shear stress) factors that facilitate epithelial-to-mesenchymal transition (EMT) and cancer metastasis [4,5]. Disseminated ovarian cancer cells show a decreased E-cadherin expression when compared to the primary tumor, a common feature of EMT [6,7]. Interestingly, these disseminated cells grow into spheroids in the peritoneal cavity that exhibit a partial EMT phenotype in which cells are positive for both E-cadherin and N-cadherin [6]. When reaching the mesothelium, these cells undergo mesenchymal-to-epithelial transition to form macroscopic metastasis [6]. The dynamic transition along the EMT spectrum allows ovarian cancer cells to demonstrate EMT plasticity, which makes them more effective in forming metastasis [7]. In contrast to most solid tumors that spread through the blood circulation, peritoneal metastases occur under a considerably different circumstance where tumor cells experience shear stress of much lower magnitude than they do in blood vessels [8]. In addition, while blood vessels are lined with endothelial cells, the peritoneal cavity is lined with mesothelial cells. Peritoneal cancer spheroids gain multiple survival advantages including the ability to adhere to the mesothelium, which is a critical determining step in ovarian cancer metastasis [2]. Several adhesion molecules have been reported to be involved in this interaction, but the role of selectins in tumor–mesothelial adhesion has not been described in most reviews [2,4,9]. Selectins are of growing interest due to their distinct binding kinetics which allows them to mediate cell-cell interaction under hydrodynamic flow, but not at static condition. They recognize specific glycan moieties as ligands. Unsurprisingly, selectins and their ligands play a key role in early events of the adhesion cascade. This review summarizes the characteristics of selectins and their ligands, and their implications in ovarian cancer metastasis. We also discuss the latest in vitro and in vivo tools employed in the study of selectins and the potential therapeutic interventions that may lead to efficacious therapies for ovarian cancer.

## 2. Role of Selectins in Ovarian Cancer Metastasis

Selectins (CD62) are glycoproteins responsible for heterophilic cell-cell interaction under hydrodynamic flow. The selectins were given their name because of their ability to bind carbohydrates via their extracellular C-type lectin domain at the N-terminal [10]. This domain is followed by an epidermal growth factor (EGF)-like domain, a variable number of consensus repeat domains, a transmembrane region, and a short cytoplasmic tail (Figure 1) [10]. The selectin family consists of three members, named after the cell-type they were first identified in: E-selectin (endothelium), P-selectin (platelets), and L-selectin (lymphocytes). The extracellular lectin domain of the three members are more than 60% identical, allowing them to selectively bind to their corresponding ligands [11]. After the successful cloning of selectins, their role in leukocyte trafficking, thrombosis, inflammation, and cancer metastasis was quickly recognized [12]. An emerging body of evidence suggests that selectins also play an active role in peritoneal carcinomatosis of ovarian cancer, summarized in Figure 2 and discussed in detail in the following sections.

### 2.1. Selectins in Peritoneal Carcinomatosis

The contribution of selectins to tumor implantation in blood-borne metastasis is well-established [13,14,15,16]. Selectins are known for their involvement in leukocyte rolling and tethering on endothelial surface [17], and it is thought that cancer cells expressing selectin ligands utilize this selectin receptor-counter receptor interaction to extravasate into blood, a key step in hematogenous metastasis [14]. This initial interaction leads to activation of integrins, which subsequently mediate firm adhesion in concert with other adhesion molecules [18]. It is presumed that similar glycan-selectin binding also takes place in non-blood-borne metastasis. Ovarian cancer metastasizes primarily into the peritoneal cavity, where tumor cells seed onto peritoneal mesothelial cells [4]. Endothelial cells and mesothelial cells are mesodermally derived cells that share several structural and functional similarities [19]. They are both flat, squamous-like cells that can change shape depending on the physiological environment and play important roles in tissue repair, immunity, homeostatic balance, and fibrinolysis regulation [20,21,22]. Importantly, these two cell types have highly similar protein expression profiles [19,23]. Although there are numerous reports of selectin expression being exclusive to leukocytes, platelets, and endothelial cells, constitutive expression of E- and P-selectin has been detected in peritoneal mesothelial cells [24,25,26].

Recently, a few studies have demonstrated rolling and adhesion of ovarian cancer cells on mesothelial cells under shear stress, indicating that a mechanism similar to tumor–endothelial interaction might be involved in adhesion of ovarian cancer cells to the peritoneal mesothelium. Oliveria-Ferrer et al. found that overexpressing c-FOS in ovarian cancer cells diminished adhesion to E-selectin and mesothelial cells under physiological shear flow conditions [27]. These c-FOS^high^ cells had lower expression of selectin ligands sialyl-Lewis^a^ (sLe^a^) and sialyl-Lewis^x^ (sLe^x^), and showed reduced metastasis in an intraperitoneal xenograft model. The potential role of other selectins was not investigated in this study. Involvement of P-selectin in rolling and adhesion of ovarian cancer cells expressing sLe^x^ on mesothelial cells was also reported recently [28]. Notably, omental tissues from patients exhibited a higher expression of P-selectin. Recent work from our lab has further highlighted the importance of P-selectin in mesothelial adhesion of ovarian cancer cells. We found that a population of metastatic cancer stem cell (M-CSC)-enriched ovarian cancer cells adhered most frequently to immobilized P-selectin among all three selectins, and blocking P-selectin had the most significant impact on tumor-mesothelial interaction under flow conditions [26]. Tumor implantation on omentum, mesenteries, and small bowels was significantly reduced in P-selectin knockout mice, even though tumor growth was comparable to wild type mice.

It is worth noting that other than ovarian cancer, there are several malignancies like pancreatic cancer, gastric cancer, and colorectal cancer in which peritoneal dissemination is commonly observed [29]. Gebauer et al. showed that pancreatic cancer cells interact with E-selectin, P-selectin, and mesothelial cells in a shear-stress dependent manner [24]. In colorectal cancer, selectin ligand sLe^a^ was found to be associated with the frequency of peritoneal carcinomatosis [30], although a causal link was not established in this study. Colon and gastric cancer cells adhere to mesothelial cells [31,32], but direct involvement of selectins in these interactions is yet to be demonstrated. An investigation into these cancers can help us understand the generic role of selectins in peritoneal carcinomatosis.

Beyond direct tumor–endothelial/tumor–mesothelial adhesion, selectins are also known to contribute to metastasis by mediating interactions between cancer cells, platelets and leukocytes [16]. Interestingly, thrombocytosis (a condition that results in high platelet counts in blood) has been known to be associated with ovarian cancer as early as 1952 [33]. In an in vivo xenograft model of ovarian cancer, reducing platelet counts using anti-platelet antibody reduced tumor growth and increased apoptosis, suggesting a key role for platelets in ovarian cancer progression as well [34]. Similar to platelets, aggregation of leukocytes is also commonly observed in the metastatic progression of ovarian cancer, and blocking some tumor–leukocyte interactions is thought to hold therapeutic potential [35,36]. It was shown in vitro that ovarian cancer cells have the ability to bind to the leukocyte homing selectin, L-selectin, albeit in a static assay [37].

### 2.2. Regulation of Selectins

All three selectin genes are located on the long arm of human chromosome 1, and contain several nearby DNA regulator elements that can control their expression [11]. While selectins have been extensively studied in platelets, leukocytes, and endothelial cells due to their longstanding roles in these cell types [16], recent studies have started to explore how this interaction is regulated in the peritoneal cavity. Carroll et al. demonstrated that alternatively activated macrophages can upregulate mesothelial expression of P-selectin via the activation of C–C chemokine receptor 5/phosphoinositide 3-kinase/Akt signaling through the secretion of macrophage inflammatory protien-1 beta (MIP-1β) [28]. Ascites from patients with high-grade serous ovarian carcinoma had higher levels of MIP-1β compared to patients with benign conditions. It is noteworthy that ascites contains high levels of numerous cytokines which are well known for inducing expression of selectins at both protein and mRNA levels. For example, vascular endothelial growth factor and tumor necrosis factor α (TNF-α) are known to induce expression of E-selectin mRNA through nuclear factor-κB [18,38]. TNF-α (and interleukin-8) can also induce proteolytic shedding of L-selectin [18]. Other cytokines in ascites that can modulate selectin expression include interleukin-1β and interleukin-10 [11,39]. In addition, shear stress can upregulate E-selectin expression in endothelial cells [40], and is a feature present in peritoneal ascites. Regulation of selectin expression in peritoneal mesothelial cells is probably the result of an interplay between these factors present in ascites, and needs to be empirically evaluated.

## 3. Selectin Ligands and Glycosyltransferases in Ovarian Cancer Metastasis

Selectins recognize specific carbohydrate moieties as their ligands. Specifically, they bind to sialylated, sulfated, fucosylated glycan sequences carried on protein or lipid backbones [12]. Glycans presented on proteins are linked to either asparagine (N-linked) or serine/threonine (O-linked) [41]. The minimum recognition motif for all three selectins are the tetrasaccharides sLe^a^ or sLe^x^ [42]. Both of these glycan structures are composed of N-acetylglucosamine, galactose, sialic acid, and fucose but differ in the bond linkage between the monosaccharides (detailed structure in Figure 3) [12]. The sequential action of glycosyl transferases N-acetylglucosaminyltransferase, galactosyltransferase, sialyl 3-galactosyltransferase, and fucosyltransferase (FUT) is responsible for the synthesis of sLe^a/x^ [41]. Aberrant expression of sLe^a^ and sLe^x^ contributes to metastasis and poor outcome in many cancers, including ovarian cancer [42,43]. Thus, there are two key players in this dysregulation of sLe^a/x^, the glycoprotein or glycolipid backbone bearing them, and the glycosyltransferases involved in their synthesis. Unsurprisingly, altered expression of glycosyltransferases is also a common feature of cancers [42]. Table 1 contains a comprehensive list of dysregulated glycosyltransferases in ovarian cancer and the ligands synthesized by them.

One class of adhesion molecules that bind to selectins with high affinity are the high molecular weight heavily glycosylated family of proteins called mucins [12]. Perhaps the most easily recognizable one in the context of ovarian cancer is mucin 16 (MUC16), also known as cancer antigen 125, which is the serum biomarker used to monitor ovarian cancer patients. MUC16 is a known selectin ligand [61], and it regulates ovarian cancer growth, tumorigenesis, and metastasis [62]. Other than MUC16, ovarian cancer cells also overexpress MUC1, MUC9, and MUC13 [45,46,63]. Among these, MUC13 enhances tumorigenesis in a xenograft mouse model. Moreover, knockdown of the glycosyltransferase responsible for synthesis of MUC1 and MUC13, N-acetylgalactosaminyltransferase 3 (GALNT3) and GALNT14, respectively, reduces migration of ovarian cancer cells. GALNT3 is also a prognostic factor for ovarian cancer. Other than mucins, important known selectin ligands in ovarian cancer include glycosaminoglycans like heparan sulfate and chondroitin sulfate, and some other glycoproteins like CD44 and CD24. Heparan sulfate proteoglycans are involved in cell adhesion and metastasis, and are overexpressed in ovarian cancer tissues [64]. Heparan sulfate 6-O sulfotransferases, enzymes involved in sulfation of heparan sulfate have also been demonstrated to be involved in angiogenesis of ovarian cancer [47]. Similar to heparan sulfate, sulfotransferases are also important for tumor-associated function of chondroitin sulfate. Carbohydrate sulfotransferases (CHSTs) are overexpressed in ovarian cancer patients, including CHST11, CHST12, CHST13, and CHST15 [49]. CHST11 is an independent prognostic factor for ovarian cancer, and chondroitin sulfate is overexpressed in the ovarian cancer extracellular matrix [65]. Another selectin ligand proteoglycan that is overexpressed in ovarian cancer is CD44, at least some isoforms [66]. Although CD44 isoforms are known to be functionally distinct, there is evidence that the glycosylation of CD44 plays a more significant role for its adhesive properties than the isoform [67]. Importantly, CD44 is known to be directly involved in peritoneal tumor implantation of ovarian cancer [68]. Lastly, CD24 is another well-known selectin ligand that is important in ovarian cancer progression. In fact, direct involvement of CD24 in P-selectin mediated tumor–mesothelial adhesion in ovarian cancer has already been demonstrated [28]. sLe^x^ expression was detected in the cell lines, but the authors did not demonstrate its presence on CD24 or investigate the possible involvement of other glycan epitopes (e.g., sLe^a^) in the interaction. Interestingly, in another study, c-FOS^high^ ovarian cancer cells that show diminished interaction with E-selectin and mesothelial cells have reduced expression of both sLe^a^ and sLe^x^ [27]. However, it was not investigated which ligand backbone carries the sLe^a/x^ antigens on these cells. Overexpression of c-FOS also resulted in altered expression of glycosyltransferases, including some β-galactoside α 2,6-sialyltransferases (ST6Gals), FUTs, GALNTs, and CHSTs. In our ovarian cancer CSC-mesothelial interaction model, we found that M-CSCs have a higher expression of FUT5 than non-metastatic (NM-) CSCs, and this results in higher expression of sLe^x^ [26]. Interestingly, although sLe^x^ was present on CD24, there was no differential expression of CD24 between M-CSCs and NM-CSCs. Insulin-like growth factor 1 receptor (IGF-1R) was found to be a sLe^x^ bearing protein which was differentially expressed between M-CSC and NM-CSC. Another observation from our study was that binding of P-selectin to IGF-1R results in phosphorylation of IGF-1R, but the downstream consequence of this activation was not evaluated. While selectins are known to mediate downstream signaling in other cancers [16], this is yet to be explored in the context of ovarian cancer.

It is interesting that despite the importance of P-selectin in peritoneal carcinomatosis, none of the studies found involvement of its primary ligand P-selectin glycoprotein ligand-1 (PSGL-1), a well-characterized ligand that is important in metastasis and platelet aggregation in other malignancies like prostate and lung cancer [69]. The possibility of PSGL-1 interaction with P-selectin in ovarian cancer was recently explored, but its expression was undetectable in cell lines [28]. Whether this is a cell line-specific phenomenon remains to be evaluated. Regardless, this demonstrates that peritoneal metastases differentially utilize selectins’ ability to recognize diverse repertoire of ligands to metastasize. However, the characterization of selectin-ligand interaction in the peritoneal cavity is far from complete, and more studies are required in order to get a comprehensive picture.

## 4. Tools for Studying Selectin Mediated Interactions

Selectins mediate early events of the cell-cell adhesion cascade pertaining to tumor cells and leukocytes under fluid flow conditions, including rolling and tethering [17]. Their ability to facilitate this is attributed to their capacity to form bonds with high association and dissociation rates, a property that requires a shear stress threshold [70,71]. From early on, there has been an emphasis on studying in vitro selectin–ligand interaction under physiologically relevant shear stress since static assays might reveal ligands that are not biologically relevant [12]. Indeed, it has been demonstrated that selectin–selectin ligand binding is mechanistically different between flow and static assays [72,73]. Most ovarian cancer studies involving selectins have utilized devices capable of generating shear stress in a controlled manner [26,27,28]. These in vitro assays incorporating shear stress have the advantage of studying its effect without added confounding factors [17]. Besides, they are amenable to manipulations such as digestion of glycan moieties with specific enzymes, activation of cells with cytokines, blocking of interaction with antibodies, and use of transfected cell lines. Devices engineered to mimic physiological shear stress in vitro broadly fall in the category of parallel-plate flow chamber, cone-and-plate flow chamber, and custom microfluidic devices. The following paragraphs will briefly outline the working principle behind these, and where appropriate their use in ovarian cancer-selectin study will be highlighted.

A typical parallel-plate flow chamber consists of a polycarbonate dish, a gasket, and a surface to coat immobilized substrate (e.g., mesothelial cells or recombinant selectin molecules) [74]. The thickness of the gasket determines the height of the chamber, and the apparatus is vacuum-sealed to ensure a constant height of the channel. Cells are introduced through an inlet into the polycarbonate dish, and the shear stress is controlled via the use of a syringe pump [74]. This setup can be viewed under a microscope to monitor and record adhesion events. Across the literature, these are the type of devices mostly commonly utilized for selectin-ligand interaction studies [75]. Indeed, most studies involving selectins in the peritoneum have utilized a commercial platform that can be considered a variant of the traditional parallel-plate flow chamber, e.g., IBIDI slides consisting of channel(s) of fixed dimensions where the immobilized substrate can be coated, and cells are perfused through an inlet connected to a syringe pump [24,27,28].

Another common device used to generate shear-stress in vitro is the cone-and-plate viscometer [74,76]. They generate shear stress with the help of a rotating cone on top of a stationary cell culture plate. The rotation ensures that cells in suspension in the culture plate encounter uniform shear stress throughout [74]. Due to the nature of the assay, they are generally not used to study selectin interaction on the immobilized substrate, but for cell-cell interactions in suspension such as tumor cells and leukocytes/platelets or selectin molecules covalently linked to beads [76,77,78]. While cone-and-plate viscometers have not been used in studying selectins specifically for ovarian cancer, they have potential utility in studying selectin-mediated tumor–leukocyte or tumor–platelet interactions.

Recent advances in microfluidics have allowed a new class of device to enter the study of selectins [79]. Microfluidics is a miniaturization technology that allows more precise control of physical parameters reconstituting the tumor microenvironment at the microscale level. It is superior in rapid screening with the use of limited primary samples from cancer patients. These devices are typically fabricated using polydimethylsiloxane (PDMS), and the structure is sealed to a glass surface. The microchannels are rectangular in shape, maintaining a constant height throughout. Shear stress in these devices is controlled via the flow rate of liquid entering through an inlet in the channel. Thus, there are many similarities between traditional parallel plate flow chambers and microfluidic devices, but the latter has few advantages due to their ease of handling, such as requiring fewer cell numbers and reagents, and flexibility of design [80]. Although the study of selectins using microfluidic devices has only started, it is important to keep in mind the potential applicability of these devices due to their ease of customization. For example, they are well poised at mimicking particular physiological aspects of the tumor microenvironment like chemical composition of ascitic fluid and co-culture of multiple cell types. It is important to draw caution to the fact that although these devices serve as excellent tools for in vitro study of selectin function in the peritoneal cavity, conclusions from them are only reliable if they implement physiologically relevant shear stress. In studies of selectin-mediated adhesion to mesothelial cells, this has ranged from 0.03 to 1.0 dyne/cm^2^ in the current literature [24,26,27,28], which is much lower than the shear stress in blood vessels (up to 200 dyne/cm^2^) [74]. This is also the case for non-selectin related ovarian cancer studies modelling shear stress in the peritoneum, reviewed recently [8].

## 5. Clinical Perspective

The involvement of selectins in various pathological conditions like inflammation and cancer has made them an attractive therapeutic target [14]. Various potential therapeutics have been designed with the aim of inhibiting selectin-glycan interactions in relation to cancer, and they include antibodies, glycomimetics, soluble forms of ligands, and small molecule inhibitors of glycosyltransferases [14,81,82]. Given the importance of selectins in cancer metastasis, it has been postulated that inhibiting selectins can prevent the metastatic spread of disease. The first selectin inhibitors developed were the carbohydrate ligand sLe^x^ that could target all three selectins, but these were unsuccessful as they had low affinity and were expensive [83]. Like sLe^x^, heparin can also inhibit all three selectins and helps to attenuate metastasis in animal models of cancer [14,84]. As an anticoagulant, low molecular weight heparin (LMWH) is recommended as the supporting antithrombotic therapy for patients with various types of cancer [85,86]. Administration of the LMWH tinzaparin can modulate drug sensitivity of platinum-resistant ovarian cancer xenografts and ovarian cancer cell lines [87,88]. There are reports that LMWHs helped in ovarian cancer patient survival during short-term administration, but did not significantly benefit patient survival after LMWH chronic administration. The inconsistent influence of LMWH on ovarian cancer in vitro and in vivo can be explained by the modulation of gene expression profile, which requires further studies [87,89]. Moreover, the abovementioned clinical trials involved patients with multiple histological cancer types. In the absence of large-scale trials, the effect of heparin on ovarian cancer patients remains unknown, including the possibility of LMWH direct influence on homeostasis and microcirculation [89,90].

Shamay et al. recently demonstrated an alternative potential therapeutic utility of selectins. They designed a nanoparticle drug carrier derived from a polysaccharide that inhibits P-selectin, allowing site-specific drug targeting and selectin inhibition at the same time [91]. Importantly, the authors found that P-selectin is overexpressed in many tumors, including ovarian cancer. While there are promising approaches to attenuate metastasis by targeting selectins, it is important to bear in mind that selectins also play a physiologically important role in the immune system. Long term inhibition of these molecules might, therefore, lead to unintended consequences. Perhaps a more feasible target would be selectin ligands since the aberrant expression of selectin ligands tend to have a cancer-specific signature.

Selectin ligands that have been targeted for therapeutic intervention in ovarian cancer include MUC1 and MUC16 [92]. MUC1 dendritic cell vaccine, that introduces cellular immune response against MUC1, was tested in Phase I/II clinical trial in the treatment of various advanced malignancies, including ovarian cancer [93]. Although further studies are required, the vaccine did not show significant clinical effect, probably due to advanced stages of cancer and a limited number of participants. Lastly, another feasible approach in clinical intervention is targeting of glycosyltransferases [81,82], which may be beneficial through their non-selectin dependent means as well. For instance, ST6Gal 1 can activate EGFR phosphorylation and render cells resistant to gefitinib [52]. Importantly, ST6Gal 1 along with β-1,4-mannosyl-glycoprotein 4-β-N-acetylglucosaminyltransferase and β-1,4-N-acetyl-galactosaminyltransferase 3 were found to be correlated with ovarian cancer cell-specific glycosylation changes [94].

## 6. Conclusions

Collective evidence suggests that selectins are vital for peritoneal carcinomatosis of ovarian cancer. They mediate the early events in adhesion of tumor cells to the mesothelium and are thus essential for peritoneal tumor implantation. Further, signaling factors responsible for upregulation of selectin expression is abundant in malignant ascites. Moving forward, it will be important to understand the role of selectins in the peritoneal microenvironment beyond tumor-mesothelial interaction. Of note, their involvement in the trafficking of platelets and leukocytes still remains to be explored.

A few studies have unveiled the glycan structures that act as selectin ligands in the peritoneal cavity. It has been observed that cell lines lacking certain ligands are unable to adhere to selectins under hydrodynamic flow [24]. Given that ovarian cancer is a highly heterogenous disease in which glycans are heavily dysregulated (see Table 1), it is necessary to evaluate the physiological relevance of these ligands in ovarian cancer progression. Wit.h the advent of new technologies such as microfluidic devices, it is going to be more feasible to conduct experiments in physiologically relevant conditions, lack of which has been a bottleneck in selectin-ligand research. Further, the regulation of clinically-relevant glycans needs to be uncovered. This, along with a more in-depth understanding of selectin functions will reveal more druggable targets that will aid in the development of efficacious therapies for ovarian cancer.

## Figures and Tables

**Figure 1 cancers-12-02238-f001:**
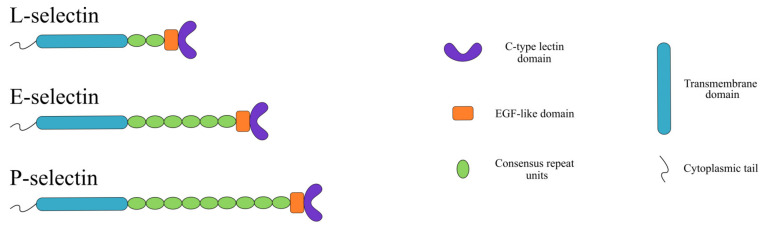
Structure of E-, P-, and L-selectin. Selectins contain a C-type lectin domain, an epidermal growth factor (EGF)-like domain, a transmembrane domain, a cytoplasmic tail, and a variable number of consensus repeats units.

**Figure 2 cancers-12-02238-f002:**
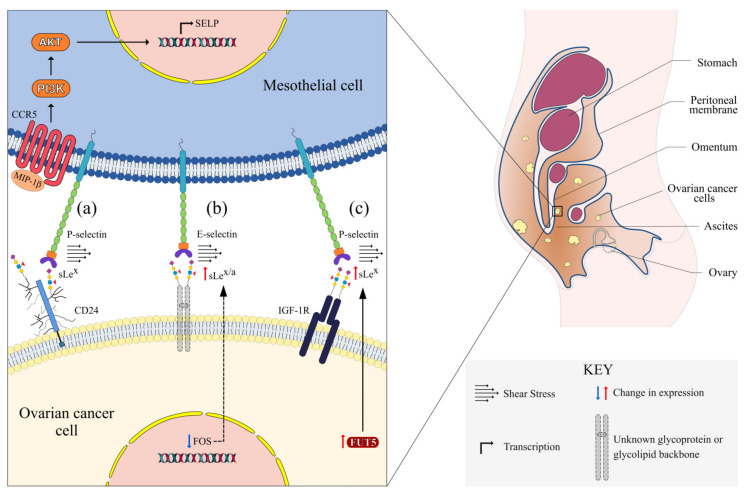
Molecular events pertinent to tumor–mesothelial adhesion mediated by selectin-sialyl-Lewis^a/x^ (sLe^a/x^) under shear stress in the peritoneal cavity. (**a**) Macrophage inflammatory protein-1β (MIP-1β) activates C-C chemokine receptor 5 (CCR5), resulting in upregulation of P-selectin gene (SELP) expression via PI3K/Akt in mesothelial cells. P-selectin interacts with sLe^x^ presented on CD24. (**b**) Downregulation of c-FOS (FOS) leads to increased sLe^a/x^ expression. E-selectin binds to sLe^a/x^ presented on an unknown glycoprotein or glycolipid backbone. (**c**) Upregulation of fucosyltransferase 5 (FUT5) results in increased sLe^x^ synthesis. sLe^x^ on insulin-like growth factor 1 receptor (IGF-1R) interacts with P-selectin.

**Figure 3 cancers-12-02238-f003:**
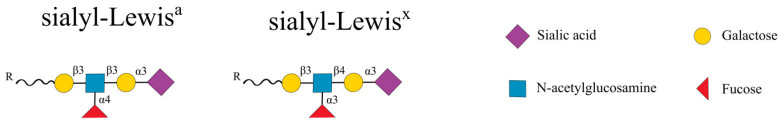
Structure of sialyl-Lewis^a^ (sLe^a^) and sialyl-Lewis^x^ (sLe^x^). Both sLe^a^ and sLe^x^ are tetrasaccharides that are sialylated by an α2,3 linkage. The fucose group is linked to N-acetylglucosamine by an α1,4 linkage for sLe^a^ and α1,3 linkage for sLe^x^, while galactose is linked to N-acetylglucosamine by a β1,3 linkage for sLe^a^ and β1,4 linkage for sLe^x^.

**Table 1 cancers-12-02238-t001:** Dysregulated glycosyltransferases in ovarian cancer.

Glycosyltransferase	Up/Down-Regulation	Target Glycan/Ligand	Phenotype	References
**N-acetylgalactosaminyltransferases (GALNTs)**				
GALNT 2	Downregulated in patient tumor	n.d.	n.d.	[44]
GALNT 3	Upregulated in patient tumor	MUC1	proliferation, invasion, migration	[45]
GALNT 6, 9	Upregulated in patient tumor	n.d.	n.d.	[44]
GALNT 12	Downregulated in c-FOS overexpression	n.d.	n.d.	[27]
GALNT 14	Upregulated in patient tumor and downregulated in c-FOS overexpression	MUC13	migration	[44,46]
**Heparan sulfate 6-O-sulfotransferases (HS6STs)**				
HS6ST 1, 2	Upregulated in patient tumor	Heparan Sulfate	angiogenesis	[47,48]
**Carbohydrate sulfotransferases** **(CHSTs)**				
CHST 11, 15	Upregulated in patient tumor and downregulated in c-FOS overexpression	Chondroitin Sulfate	n.d.	[27,49]
CHST 12, 13	Upregulated in patient tumor	Chondroitin Sulfate	n.d.	[49]
**Fucosyltransferases(FUTs)**				
FUT 1	Upregulated in patient tumor	Lewis^y^	proliferation, migration, invasion, 5-fluorouracil resistance	[50,51]
FUT 5	Upregulated in metastatic cancer stem cells and patient tumor	sialyl-Lewis^x^ on IGF-1R	cell-cell adhesion	[26]
FUT 11	Downregulated in c-FOS overexpression	n.d.	n.d.	[27]
**β-galactoside α2-6-sialyltransferase 1 (ST6Gal 1)**	Upregulated in patient tumor and downregulated in c-FOS overexpression	EGFR, β1 Integrin	cisplatin resistance, gefitinib resistance, cell-ECM adhesion, invasion, migration, shorter recurrence free survival	[27,52,53,54,55,56]
**β-galactoside α2-3-sialyltransferase 1 (ST3Gals)**				
ST3Gal 1	Upregulated in patient tumor	n.d.	proliferation, invasion, migration, paclitaxel resistance	[54,57]
ST3Gal 3,4	Upregulated and downregulated (conflicting reports) in patient tumor and upregulated in metastatic cancer stem cells	n.d.	taxol resistance (ST3Gal 3 upregulation)	[26,54,58]
ST3Gal 6	Downregulated in patient tumor	n.d.	n.d.	[54]
**β-1, 4-Galactosyltransferase 4** **(B4GalT 4)**	Upregulated in metastatic cancer stem cells and patient tumor	n.d.	platinum resistance, invasion, migration	[26,59,60]

n.d.: not determined; MUC: mucin; IGF-1R: insulin-like growth factor 1 receptor; EGFR: epidermal growth factor receptor.

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
