# Peer review of "Selectins: An Important Family of Glycan-Binding Cell Adhesion Molecules in Ovarian Cancer"

_cancers, 2020, doi:10.3390/cancers12082238_

Round 1
Reviewer 1 Report
The authors explained the function of selectin in ovarian cancer. The issue is important and valuable in glycobiology and cancer biology fields. However, the issue was not reported as a review article in recent 10 years, except only one article in the Glycoconjugate journal, 10.1007/s10719-020-09912-4. Thus, I think this manuscript has merit to deal with a review.
The Figure in the manuscript should be original. If the Figures were reused from other papers, they should show the reference and permission of the previous reports. The issue should be confirmed.
In Figure 1, the authors did not show the L-selectin structure. It should be included.
Author Response
Point 1: The authors explained the function of selectin in ovarian cancer. The issue is important and valuable in glycobiology and cancer biology fields. However, the issue was not reported as a review article in recent 10 years, except only one article in the Glycoconjugate journal, 10.1007/s10719-020-09912-4. Thus, I think this manuscript has merit to deal with a review.
Response 1: Thank you for the comment. We are pleased that the reviewer agreed that the review is important and valuable.
Point 2: The Figure in the manuscript should be original. If the Figures were reused from other papers, they should show the reference and permission of the previous reports. The issue should be confirmed.
Response 2: Figure 2 “the diagram showing fluid in the abdomen” is adopted from Wikimedia Commons, which is free to be used by anyone for any purpose. However, to avoid copyright issue, we have now made our own diagram. The other figures are all original.
Point 3: In Figure 1, the authors did not show the L-selectin structure. It should be included.
Response 3: In response to the reviewer’s comment, the L-selectin structure is now included.

Reviewer 2 Report
The review is timely and interesting. What I miss is a discussion why selectins are of interest. Selectins are the first point of call for leukocytes adhering to endothelial cells at the site of inflammation. This initial loose adhesion is followed by a tight adhesion mediated e. g. by integrins and cell adhesion moleculesof the immunoglobulin superfamily. Therefore selectins are of importance in distant metastases.
Furthermore it is not mentioned that both endothelia covering the blood vessels and mesothelial cell covering the peritoneal cavity are both derivatives of the mesoderm, hence both are msothelial cells. They are structurally, functionally and embryologically related cells. No wonder they share the same adhesion molecules.
Cells metastasising from solid tumors which metastasize to distant organs must undergo epithelial to mesenchymal transition (EMT). This EMT may be limited in intraperitoneal metastasis as ovarian cancer cells must only lose themselves from the primary tumour and are probably not required to acquire the ability to adhere to endothelial cells initially as they can grow in spheroids in the ascites which gives them the opportunity ot acquire the ability to adhere to the mesothelium as an intermediate stepp. As cancer cells in the bloodstream have a very short half life this way of adaption is not open to them.
Author Response
Point 1: The review is timely and interesting. What I miss is a discussion why selectins are of interest. Selectins are the first point of call for leukocytes adhering to endothelial cells at the site of inflammation. This initial loose adhesion is followed by a tight adhesion mediated e. g. by integrins and cell adhesion molecules of the immunoglobulin superfamily. Therefore selectins are of importance in distant metastases.
Response 1: We thank the reviewer for the comments. As suggested, the importance of selectins in ovarian cancer progression is further highlighted in “1. Introduction” (p. 2, lines 49-53), and the subsequent integrin activation in “2.1. Selectins in peritoneal carcinomatosis” (p. 2-3, lines 80-81).
Point 2: Furthermore it is not mentioned that both endothelia covering the blood vessels and mesothelial cell covering the peritoneal cavity are both derivatives of the mesoderm, hence both are msothelial cells. They are structurally, functionally and embryologically related cells. No wonder they share the same adhesion molecules.
Response 2: It is correct that both endothelia and mesothelial are derived from mesoderm and their structural, functional, and embryonic similarities are now described in “2.1. Selectins in peritoneal carcinomatosis” (p. 3, lines 84-88).
Point 3: Cells metastasising from solid tumors which metastasize to distant organs must undergo epithelial to mesenchymal transition (EMT). This EMT may be limited in intraperitoneal metastasis as ovarian cancer cells must only lose themselves from the primary tumour and are probably not required to acquire the ability to adhere to endothelial cells initially as they can grow in spheroids in the ascites which gives them the opportunity ot acquire the ability to adhere to the mesothelium as an intermediate stepp. As cancer cells in the bloodstream have a very short half life this way of adaption is not open to them.
Response 3: It is correct that EMT is an essential process in peritoneal metastasis, and we have now described its involvement in “1. Introduction” (p. 1, lines 34-41 and p. 2, lines 45-46).

Round 2
Reviewer 2 Report
Revisions are well done.